# Cathodic Protection of A Container Ship Using A Detailed BEM Model

**Dimitrios T. Kalovelonis [1], Dimitrios C. Rodopoulos [1], Theodoros V. Gortsas [2],**
**Demosthenes Polyzos [1] and Stephanos V. Tsinopoulos [2,\*]**

[1]  Department of Mechanical Engineering & Aeronautics, University of Patras, 26504 Patras, Greece;
     mead6346@upnet.gr (D.T.K.); d.rodopoulos@upnet.gr (D.C.R.); polyzos@mech.upatras.gr (D.P.)
[2]  Department of Mechanical Engineering, University of Peloponnese, 26334 Patras, Greece; thgkortsas@uop.gr
\*   Correspondence: stsinop@uop.gr

**Abstract:** In the present work, an impressed current cathodic protection (ICCP) system for the protection against corrosion of a 399-m-length container ship throughout its service life is designed. The study is carried out with the aid of a boundary element method code, accelerated by an adaptive cross approximation scheme, utilizing a detailed large-scale model. The exact geometry of the ship, the progressive damage of the coating system, and the dynamic state during the cruise of the ship are the main parameters taken into consideration in modelling. The main objective of the design process is to minimize the electric power, delivered by the ICCP system, determining the optimal number and location of the installed inert anodes to accomplish the absolute minimum protection potential on the immersed steel surfaces of the ship and, simultaneously, avoid overprotection. Performing an extensive parametric study, a six-zone ICCP system is proposed, consisting of 10 anodes at the hull and four identical anodes at each of four thrusters.

**Keywords:** cathodic protection; ICCP system; boundary element method (BEM); large-scale problems; adaptive cross approximation; ACA/BEM

## 1. Introduction

The steel structure of a ship that is exposed to seawater is naturally prone to corrode. Use of coating systems is standard practice for corrosion control. However, mechanical wear, biological attack, and aging occur throughout the service time of a ship. As a result, the coating systems breakdown, losing incrementally their ability to prevent susceptibility to corrosion. The combination of coating and impressed current cathodic protection (ICCP) systems ensures adequate corrosion protection [1–12] throughout the service life of a ship. In modern ICCP systems, the delivered current is regulated in varying conditions with the aid of automated control systems.

Seawater temperature, pH, salinity, and oxygen concentration influence corrosion initiation. Furthermore, the galvanic coupling between dissimilar materials, turbulent flow caused by propellers, dynamic conditions during the cruise of the ship, and biofouling all affect the corrosion rate [5]. The design of an ICCP system is dependent on the polarization characteristics of the exposed metals and alloys, the chemical and electrical properties of seawater, the geometry of the immersed surfaces, and the service life and the operational aspects of the ship.

A well-established and powerful numerical tool for analyzing cathodic protection problems is the boundary element method (BEM) [4–21]. One remarkable advantage it offers is the reduction of the dimensionality of the problem by one. Thus, although the electrochemical process occurs at the metallic surface and the surrounding electrolyte volume, cathodic protection (CP) problems can be solved by discretizing only the under protection metallic surface. In the last few decades, BEM has

been extensively used to solve sacrificial CP and ICCP problems mainly in ships [4–18], offshore and subsea structures [19,20], buried pipelines, underground structures, and steel-reinforced concrete [21], as well as optimization problems regarding the optimum placement of the anodes [8].

However, despite its advantages, the application of conventional BEM to large-scale problems suffers from very time-consuming computations and high demands for computer memory capacity. On the other hand, the design of an efficient ICCP system demands accurate modelling of the exact geometry of complex metallic structures. The use of hierarchical matrices (HM) and adaptive cross approximation (ACA) techniques, accelerate the computation of matrix [A] drastically and also reduce the memory requirements [20].

In this work, an ICCP system of a container ship is designed, with the aid of the ACA/BEM code, proposed in a previous work of the present co-authors [20], utilizing a detailed large-scale model. The problem is modelled, taking into consideration the exact geometry of the ship, the progressive damage of the coating system in various regions and the dynamic conditions during cruise. The main objectives of the design process are the following: (i) the calculation of the optimal number and location of the installed inert anodes to achieve the required absolute minimum protection potential on the immersed steel surfaces of the ship; (ii) the determination of electrical insulation needed, around the IC anodes, to avoid overprotection; and (iii) the estimation of the location of the reference electrodes used as sensors to adjust the delivered current.

## 2. Governing Equations of an ICCP Problem for Immersed Structures in Semi-Infinite Electrolytes and ACA/BEM

### 2.1. Governing Equations

Consider a structure of surface $S$, immersed in a semi-infinite homogenous electrolyte $\Omega$ of boundary $S_0$ and electrical conductivity $\sigma$, enclosed by an artificial boundary $S_\infty$, as shown in Figure 1. The surface $S$ is divided into a cathode $S_c$, an anode $S_a$, and an electrically insulated $S_p$. In the absence of sources and sinks, conservation of charge implies [22]

$$\nabla \cdot \mathbf{J} = 0, \tag{1}$$

where $\mathbf{J}$ is the current density vector and $\nabla$ is the nabla operator.

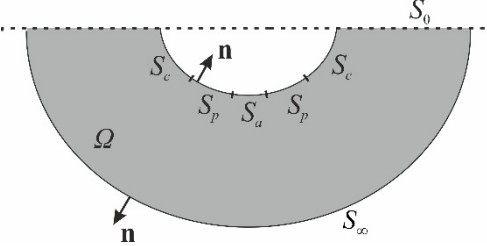

**Figure 1.** Illustration of an impressed current cathodic protection (ICCP) problem of a structure immersed in a semi-infinitely extended electrolyte.

At steady-state conditions, the current density is proportional to the electric potential gradient (Ohm's law), i.e.,

$$\mathbf{J} = \sigma \mathbf{E} = -\sigma \nabla \varphi, \tag{2}$$

where $\sigma$ is the electric conductivity of the electrolyte, $\varphi$ is the electric potential, and $\mathbf{E}$ is the electric field.

Under the assumption of homogenous electrolyte, Equation (1), in conjunction with Equation (2), reduces to the Laplace equation

$$\nabla^2 \varphi = 0. \tag{3}$$

The current density $i$ at a surface with a normal unit vector $\mathbf{n}$, in terms of $\varphi$ is expressed as follows:

$$i = \mathbf{n} \cdot \mathbf{J} = -\sigma \mathbf{n} \cdot \nabla\varphi = -\sigma\partial_n\varphi, \tag{4}$$

where $\partial_n\varphi = \mathbf{n} \cdot \nabla\varphi$ is the directional derivative concerning the normal unit vector $\mathbf{n}$, defined at all surfaces (Figure 1).

The integral representation of Equation (3) at a field point $\mathbf{x}$, reads [20]

$$c(\mathbf{x})\varphi(\mathbf{x}) + \int_{S\cup S_\infty} \partial_n G(\mathbf{x}, \mathbf{y})\varphi(\mathbf{y})dS_y = \int_{S\cup S_\infty} G(\mathbf{x}, \mathbf{y})\partial_n\varphi(\mathbf{y})dS_y, \tag{5}$$

where $\mathbf{y}$ is a source point, lying on the boundary $(S\cup S_\infty)$; $c(\mathbf{x})$ is a jump coefficient, taking the values 0.5 or 1 when $\mathbf{x}$ is on the boundary $(S\cup S_\infty)$ or 1 inside the electrolyte $\Omega$, respectively; and $G(\mathbf{x,y})$ is the modified 3D half-space fundamental solution, given from the following expression [20]:

$$\begin{aligned} G(\mathbf{x}, \mathbf{y}) &= \tfrac{1}{4\pi r} + \tfrac{1}{4\pi r'} \\ \partial_n G(\mathbf{x}, \mathbf{y}) &= -\tfrac{1}{4\pi r^2}\mathbf{n} \cdot \nabla r' - \tfrac{1}{4\pi r r'^2}\mathbf{n} \cdot \nabla r' \quad, \\ r' &= |\mathbf{y} - \mathbf{x}'| \end{aligned} \tag{6}$$

where $\mathbf{x}'$ is the reflection of the field point $\mathbf{x}$ with respect to the semi-infinite free surface $S_0$ of the electrolyte. The advantage of utilizing the modified half-space fundamental solution Equation (6), instead of the 3D full-space corresponding solution, is that the discretization of the semi-infinite boundary $S_0$ is avoided.

At the boundaries $S$ and $S_\infty$, the following boundary conditions are imposed. At perfectly electric insulated boundary $S_p$ and at the artificial boundary $S_\infty$, the current density vanishes. At the cathode $S_c$, a nonlinear Robin boundary condition between potential and current density is imposed, known as a polarization curve, which is usually derived experimentally, taking into account the real conditions of the galvanic cell. At the anode boundary $S_a$, a known current density is imposed. Consequently, the above-described boundary conditions read

$$i(\mathbf{x}) = -\sigma\partial_n\varphi(\mathbf{x}) = 0, \ \mathbf{x} \in S_p, \tag{7}$$

$$i_c(\mathbf{x}) = f(\varphi_c(\mathbf{x})), \mathbf{x} \in S_c, \tag{8}$$

$$i_a(\mathbf{x}) = -\sigma\partial_n\varphi(\mathbf{x}) = J_0, \ \mathbf{x} \in S_a, \tag{9}$$

$$i(\mathbf{x}) = -\sigma\partial_n\varphi(\mathbf{x}) = 0, \mathbf{x} \in S_\infty, \tag{10}$$

with $f$ being the polarization curve at the cathode $S_c$. Boundary condition Equation (10), imposed at $S_\infty$, implies that Gauss law, which states that the total electric flux through any closed surface is equal to the total charge enclosed by the surface, is fulfilled.

## 2.2. ACA/BEM

The above-described CP problem can be solved efficiently with BEM. The advantage of the reduction of problem's dimensionality by one makes BEM an ideal tool for dealing with problems of infinite or semi-infinite domains [4–21].

According to BEM, the boundaries $S$ and $S_\infty$ are discretized into quadrilateral or/and triangular surface elements. Each element has a certain number of geometrical and functional nodes. The entire set of the geometrical nodes form the mesh of the discretization. On the surface of each element, the geometry is approximated via polynomial shape functions of specific order and the coordinates of its geometrical nodes, i.e., the continuous surface is described approximately by a finite number of geometrical nodes in 3D space. Similarly, on the surface of each element, both unknown and known (by the imposed boundary conditions) fields, potential $\varphi$, and current density $i$ are assumed to vary

as polynomial functions of specific order and are approximated via interpolation functions, and the corresponding field values at the functional nodes, i.e., the infinite Degrees of Freedom (DoFs) of the continuous medium, are interpolated by a finite number of nodal values.

Then, the integral Equation (5) is transformed in a discretized form and is collocated for each functional node. The integrals over each surface element are either regular or singular and are evaluated numerically, employing gauss quadrature for the former and special integration techniques for the latter. Assembling the discretized integral Equations (5) of all functional nodes, one obtains the following system of algebraic equations:

$$[\mathbf{H}] \cdot \{\boldsymbol{\phi}\} = [\mathbf{G}] \cdot \{\mathbf{i}\}, \tag{11}$$

where the elements of matrices [**H**] and [**G**] contain a summation of integrals, while the vectors $\{\boldsymbol{\phi}\}$ and $\{\mathbf{i}\}$ contain all the known and unknown values of potential and current density, respectively, at functional nodes.

At cathode surface $S_c$, due to the nonlinear nature of the boundary conditions, a Newton–Raphson iterative procedure is applied. According to this scheme, the boundary conditions Equation (8) obtain the following incremental form:

$$\partial_n \varphi_c^k(\mathbf{x}) = \frac{1}{\sigma} f\left(\varphi_c^{k-1}(\mathbf{x})\right) + \frac{1}{\sigma} \frac{\partial f\left(\varphi_c^{k-1}(\mathbf{x})\right)}{\partial \varphi} \Delta \varphi_c^k(\mathbf{x}), \mathbf{x} \in S_c$$
$$\varphi_c^k(\mathbf{x}) = \varphi_c^{k-1}(\mathbf{x}) + \Delta \varphi_c^k(\mathbf{x}) \tag{12}$$

where $k$ is the iteration step.

Rearranging Equation (11), with the aid of Equations (7), (9), (10), and (12), one obtains the following linear algebraic system of equations, for each iteration $k$:

$$\left[\mathbf{A}^{k-1}\right] \cdot \left\{\mathbf{x}^k\right\} = \left\{\mathbf{B}^{k-1}\right\}. \tag{13}$$

Finally, the linear algebraic system of Equations (13) is solved, for each iteration $k$, by means of the iterative GMRES solver.

As soon as, the boundary value problem has been solved and potential and current density have been calculated at the functional nodes, the potential at a point **x** located inside the electrolyte can be easily obtained by integral Equation (5), with $c(\mathbf{x}) = 1$.

The use of HM and ACA techniques accelerates the computation of matrix [**A**] drastically and also reduces the memory requirements. According to ACA/BEM, the matrix [**A**] is organized into a hierarchical structure of blocks, depending on the geometry of the problem. Applying a geometrical criterion, the blocks are characterized either as non-admissible, where the ACA algorithm is inefficient, and thus the conventional BEM is employed, or admissible, where ACA is applied, and thus, only a small number of their rows and columns are calculated. Each admissible block is approximated with a low-rank matrix via a small summation of vector products, formed by the previously calculated rows and columns. This low-rank matrix format, in conjunction with an iterative solver, like GMRES, leads to a significant reduction in memory requirements and CPU time, due to the acceleration of the matrix-vector multiplication

Details of the above described in brief, ACA/BEM, one can find in Rodopoulos et al. [20].

## 3. ICCP System Design of an 18270 TEU Capacity Container Ship

In this section, an ICCP system of an 18270 20-foot equivalent unit (TEU) capacity container ship is designed. To this aim, the ICCP problem is solved numerically, with the aid of the above-described ACA/BEM code, utilizing a detailed large-scale model.

The main objective of the study is to calculate the optimal number and location of the installed inert anodes to achieve the required absolute minimum protection potential, delivering the minimum

current. Furthermore, the electrical insulation needed around the IC anodes to avoid overprotection and the locations of the reference electrodes are determined.

### 3.1. Description of the Container Ship

The analyzed container ship has a length 399 m, a width 59 m, and a fully loaded waterline depth 24.5 m, as depicted in Figure 2. The wetted area of the hull surface is 32,690 m$^2$. The main propeller and the rudder have an area of a 146 m$^2$ and 235 m$^2$, respectively. The ship has four thrusters, each consisting of two identical maneuvering propellers attached to a common shaft located in a tube, as shown in Figure 3a. The total area of the four thruster surfaces is 1045 m$^2$. Consequently, the total immersed surface of the vessel is 34,116 m$^2$.

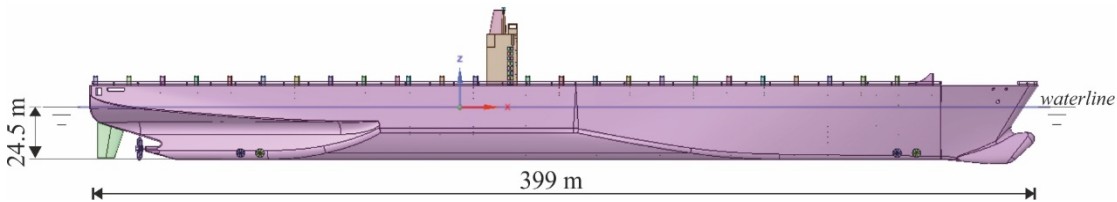

**Figure 2.** Side view of the analyzed 18270 20-foot equivalent unit (TEU) container ship.

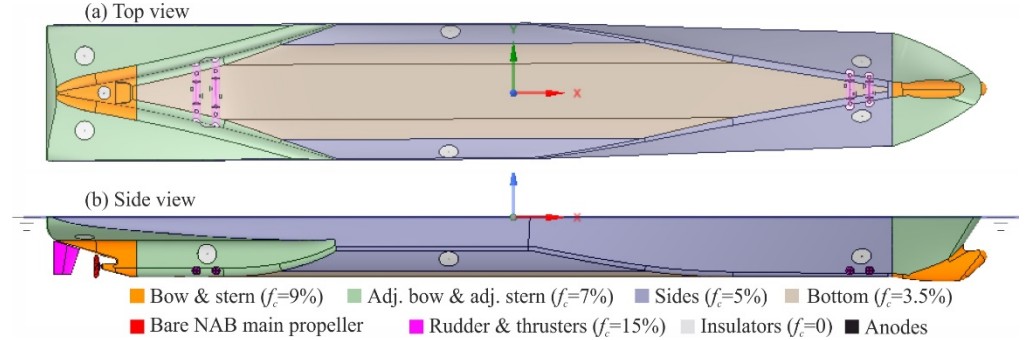

**Figure 3.** Coating damage to various regions of the ship.

The surfaces of the vessel are made of steel CSA G40.8 grade 8, except the main propeller, which is nickel-aluminum-bronze (NAB) alloy. The steel surfaces are coated, with marine epoxy paint with a three-year design life and a total nominal dry film thickness 350 μm, classified to the Category III, according to Det Norske Veritas (DNV) [23]. In general, NAB exhibits excellent corrosion resistance, appearing corrosion rates of 0.015–0.05 mm y$^{-1}$ [24] and consequently, no protection is needed.

### 3.2. Polarization Characteristics of the Coated Surfaces

Protective coatings break down during the operational time of a ship, failing incrementally to isolate the exposed steel surface from its corrosive marine environment. As a result, the delivered current by the ICCP system to maintain protection is increased over time. Furthermore, the coating degradation varies over the hull surface due to different operational conditions.

According to DNV [23], the coating damage is modelled adopting an effective area, calculated as a percentage of the total immersed area by a coating breakdown factor $f_c$. The factor $f_c$ takes values in the range [0, 1]. For $f_c = 0$, the coating is a perfect electrical insulator, while for $f_c = 1$ no effective coating exists (bare metal). For a Category III coating with a three-year design life, the breakdown factor results in an $f_c = 5.6\%$ [23].

In the present work, the factor $f_c$ is used to obtain the polarization curve of a coated surface from the corresponding uncoated surface, reducing the current densities proportionally by $f_c$, for the same potential. The above-calculated value of $f_c = 5.6\%$ is used as a total weighted average of the coating degradation varied over the hull surface. According to observed coating damage during

maintenance of commercial ships [1], the coating degradation is modelled, grouping the immersed hull surface into six regions of constant damage, as shown in Figure 3. The areas of each region are bow—710 m$^2$, adjacent bow—1743 m$^2$, sides—14,231 m$^2$, bottom—9228 m$^2$, adjacent stern—5780 m$^2$, and stern—998 m$^2$. At the end of the coating's design life, the damage is assumed to be 9% at the bow and stern regions, 7% at the adjacent bow and adjacent stern regions, 5% at the sides region, and 3.5% at the bottom region. Finally, increased damage of 15% is assumed to the rudder and thrusters surfaces due to a high degree of turbulence. The weighted average of the above breakdown factor values is 5.61%.

Furthermore, while a ship is in motion, an increased amount of current density must be delivered to the immersed surfaces of the hull due to raised oxygen concentration, which increases the corrosion rates. According to Lucas et al. [2], the dynamic state may increase the current demand by 3–5 times with respect to the static state. In the present study, an additional factor $f_d$, is used, in a similar way with the $f_c$, to take into account the above-mentioned increased current density demand in dynamic conditions for the steel surfaces.

Thus, the polarization curves of coated steel surfaces in seawater, are given by the non-linear relation [12]

$$i(\varphi, f_c) = f_d f_c \left[ i_{corr} e^{(\varphi - \varphi_{corr})/b_a} - i_{corr} e^{-(\varphi - \varphi_{corr})/b_c} \right], \tag{14}$$

where $\varphi$ and $i$ is expressed in V (vs. Ag/AgCl/seawater reference electrode, RE) and A/m$^2$, respectively, and $i_{corr} = 1.63 \, \text{mA/m}^2$, $\varphi_{corr} = -0.656 \, \text{V}$ (vs. Ag/AgCl/seawater), $b_a = 0.0434 \, \text{V/dec}$, and $b_c = 0.0434 \, \text{V/dec}$ are constants measured in static conditions. In the present study, the value of $f_d = 4$ has been chosen.

The polarization curve of the bare NAB alloy in seawater, at the rotation speed of 800 RPM, is given by the following relation ([25]):

$$i(\varphi, f_c) = i_{corr} e^{(\varphi - \varphi_{corr})/b_a} - i_{corr} e^{-(\varphi - \varphi_{corr})/b_c}, \tag{15}$$

where $\varphi$ and $i$ is expressed in V (vs. Ag/AgCl/seawater RE) and A/m$^2$, respectively, $i_{corr} = 0.1 \, \text{mA/m}^2$, $\varphi_{corr} = -0.33 \, \text{V}$ (vs. Ag/AgCl/seawater), $b_a = 0.0651 \, \text{V/dec}$, and $b_c = 0.0651 \, \text{V/dec}$.

In Figure 4, the polarization curves of coated steel in seawater, for all adopted coating degradation conditions and the corresponding bare conditions, as obtained by Equation (14), are plotted. Similarly, in Figure 5, the adopted corresponding polarization curve for the NAB alloy is shown.

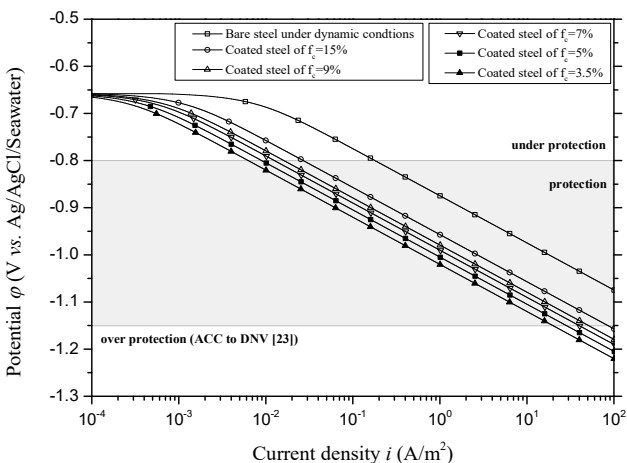

**Figure 4.** Polarization curves (only the cathodic branch) of coated steel in seawater for all adopted coating degradation conditions.

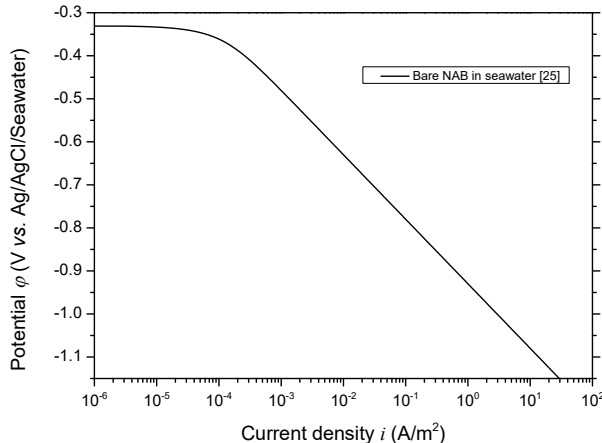

**Figure 5.** Polarization curve (only the cathodic branch) of bare nickel-aluminum-bronze (NAB) alloy in seawater.

From Figure 4, one can observe that, for bare steel, the corresponding protection potential of −0.8 V (vs. Ag/AgCl/seawater RE) [23] current density demand is 181 mA/m². As already explained, the corresponding demand for various coated conditions is reduced proportionally by their factor $f_c$, i.e., 27.2 mA/m² for $f_c$ = 15%, 16.3 mA/m² for $f_c$ = 9%, 12.7 mA/m² for $f_c$ = 7%, 9.1 mA/m² for $f_c$ = 5%, and 6.3 mA/m² for $f_c$ = 3.5%.

### 3.3. Configuration of the ICCP System

Performing an extensive parametric study, where many ICCP configurations were evaluated, differing in the number and location of the installed anodes, as well as in the amount of the current they deliver, a six-zone ICCP system for the above-described container ship, as shown in Figures 3 and 6, is proposed.

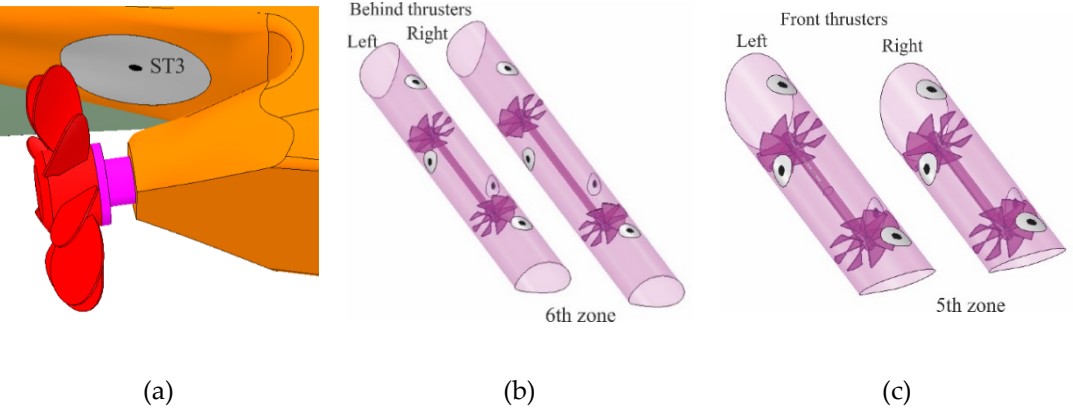

(a)　　　　　　　　　　　　　　(b)　　　　　　　　　　　　　　(c)

**Figure 6.** Details of installed anodes. (**a**) ST3 anode of the fourth zone at the stern (**b**) The sixth zone behind the thrusters and (**c**) the fifth zone in front of the thrusters. Black and grey areas represent anode and insulated areas, respectively.

In the first zone, two circular inert anodes (FR) are installed symmetrically in the front area, very close to the front pair of thrusters in a distance of 50 m from the bow (foremost part of the ship). In the second zone, two circular inert anodes (CE) are mounded symmetrically in the central area, in a distance of 26 m from the center of the ship to the stern direction. In the third zone, two circular inert anodes (BE) are installed symmetrically in the behind area, very close to the behind pair of thrusters at a distance of 66 m from the stern (the aft-most part of the ship). In the fourth zone, two circular inert anodes (ST) are installed symmetrically in the stern area in a distance of 15 m from the

stern. In this zone, an additional circular inert anode (ST3) is mounded near to the main propeller, as shown in Figure 6a. The fifth and sixth zones concern the pair of front and back thrusters, respectively. Four identical circular inert anodes are installed in each thruster, as shown in Figure 6b,c.

Table 1 lists the current *I* supplied to the installed inert anodes of the six-zone ICCP system and the total current per zone and its corresponding percentage. From Table 1, one can observe that the portion of current consumed at the stern areas is 22.8%, at the remaining hull is 70.6%, and at the four thrusters is 6.6%. The total delivered current is 2518.8 A.

**Table 1.** Current *I* (A) supplied by the proposed ICCP system configuration.

| ICCP zone | | | Current *I* (A) Per Anode | Current *I* (A) Per Zone | Current *I* Percentage (%) Per Zone |
|---|---|---|---|---|---|
| No. | Anode Title | Anode No | | | |
| 1 | Front, FR | 2 | 346.8 | 693.6 | 27.54 |
| 2 | Center, CE | 2 | 345.9 | 691.8 | 27.47 |
| 3 | Behind, BE | 2 | 196.5 | 393.0 | 15.60 |
| 4 | Stern, ST | 2 | 224.5 | 575.0 | 22.83 |
| | ST3 | 1 | 126.0 | | |
| 5 | Tube Front-R | 4 | 4.3 | 52.0 | 2.06 |
| | Tube Front-L | 4 | 8.7 | | |
| 6 | Tube Behind-R | 4 | 14.2 | 113.4 | 4.50 |
| | Tube Behind-L | 4 | 14.2 | | |
| | Total | 26 | - | 2518.8 | 100.0 |

In order to avoid cathodic disbonding caused by overprotection, at adjacent areas of the installed anodes, a circular area around each anode is perfect electrically insulated. At the proposed ICCP configuration, a radius of 4 m and 1 m for the anodes mounded on the hull and thrusters, respectively, has been considered. As an exception, the radius of the insulated area around the anode ST3 (Figure 6a) is of 3 m due to geometric constraints.

### 3.4. BEM Model

The electric conductivity of seawater, with a salinity 35% at a temperature of 12 °C, has been taken as $\sigma = 4$ S/m [23]. As mentioned in Section 2.1, the system structure electrolyte must be embedded in an artificial boundary, where the zero current density boundary condition Equation (10) is imposed. Taking into account that the half-space fundamental solution is employed to avoid the discretization of the infinite free surface of seawater, as well as the longitudinal geometry of the ship, a semi-cylindrical artificial boundary of a length 700 m and a diameter 300 m has been adopted, as shown in Figure 7.

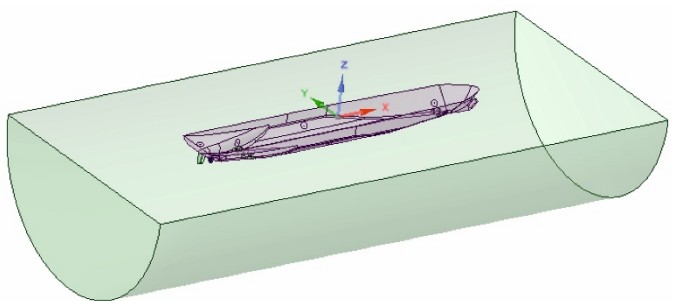

**Figure 7.** The semi-cylindrical artificial boundary, where the zero current density boundary condition is imposed.

In a BEM model, only the surface of the seawater touching the outer surface of the ship, as well as the surface of the semi-cylinder, is required to be discretized. In the present model, they are discretized by eight-noded quadratic quadrilateral and six-noded quadratic triangular elements, while for the

interpolation of both potential and current density fields, constant elements have been employed, resulting to a problem with 203941 DoFs. Figure 8 depicts some discretization details.

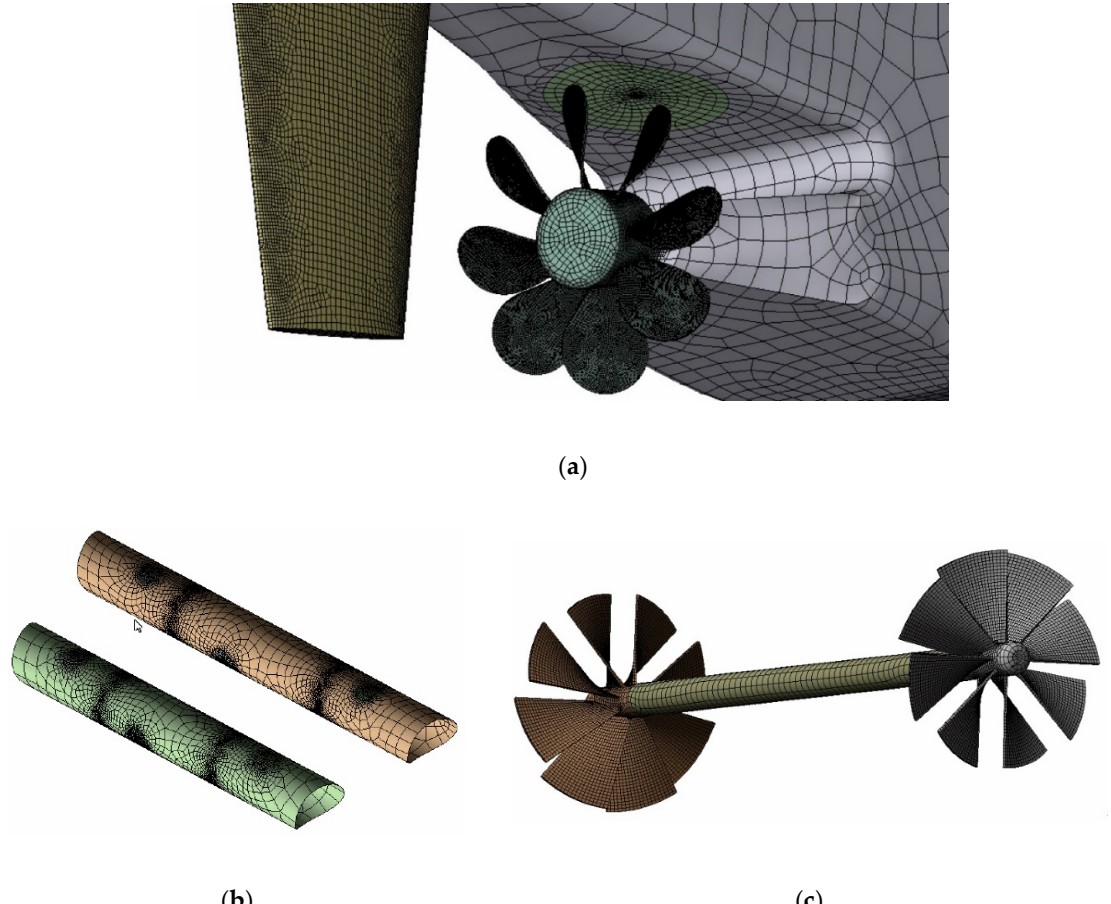

(**a**)

(**b**)          (**c**)

**Figure 8.** Details of boundary element discretization. (**a**) Stern region; (**b**) thruster tubes, (**c**) thruster propellers shaft.

In order to test the solution convergence, another model, with a denser discretization of 446,383 DoFs, has also been solved. Furthermore, in this model, a larger semi-cylindrical artificial boundary with a length of 2800 m and a diameter of 1200, has been adopted in order to validate that the chosen values of 700 m and 300 m, respectively, do not have any effect on the numerical results. The excellent agreement in the results, with the maximum absolute difference on both electrical potential and current density being below 1%, verifies the numerical convergence of the two models.

*3.5. Numerical Results and Discussion*

The calculated electric potential distribution at the immersed surface of the ship is depicted in the contour plots of Figures 9–14. More specifically, Figures 9 and 10 depict the potential distribution at the steel hull; Figure 11 at the steel rudder; Figures 12 and 13 at the steel tubes and maneuvering propellers of the thrusters, respectively; and Figure 14 shows the electric potential distribution at the NAB main propeller.

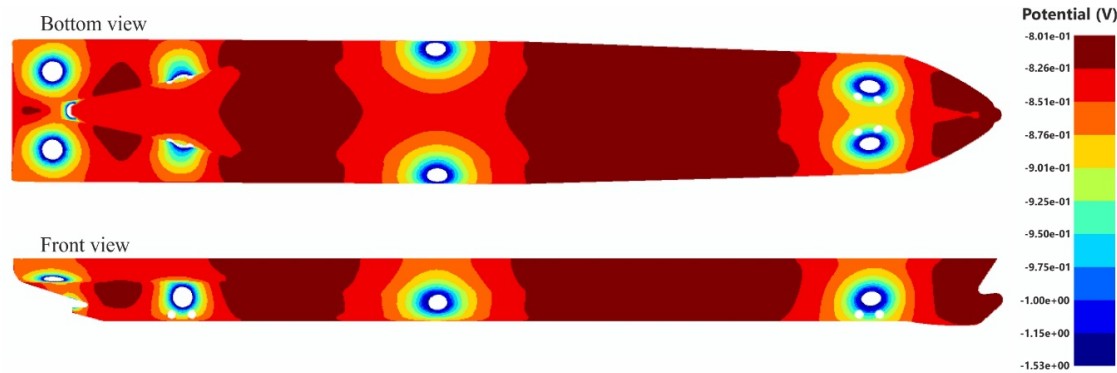

**Figure 9.** Potential distribution φ (V vs Ag/AgCl/seawater) at the steel hull.

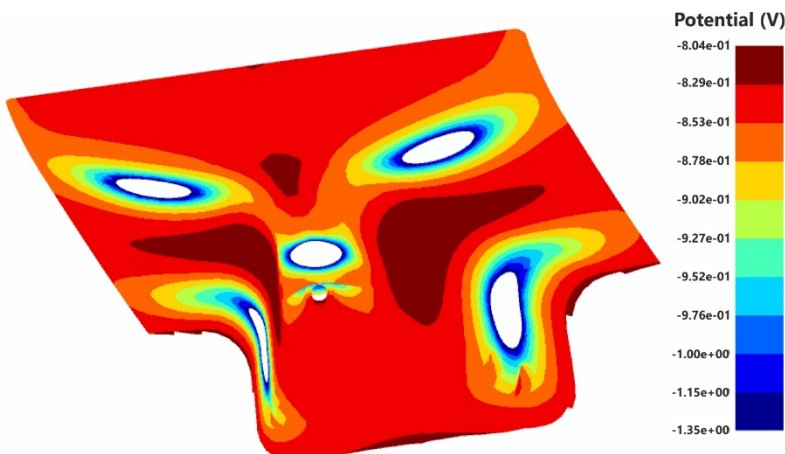

**Figure 10.** Potential distribution φ (V vs Ag/AgCl/seawater) at the stern area of steel hull.

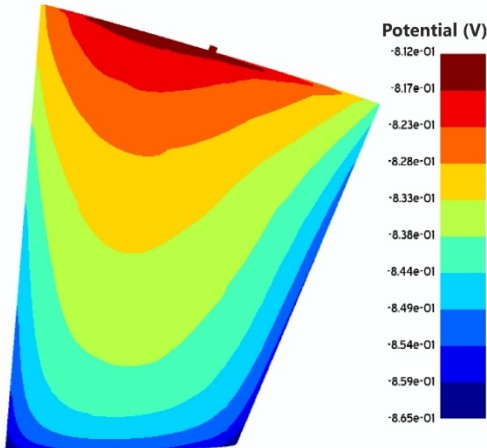

**Figure 11.** Potential distribution φ (V vs Ag/AgCl/seawater) at the steel rudder.

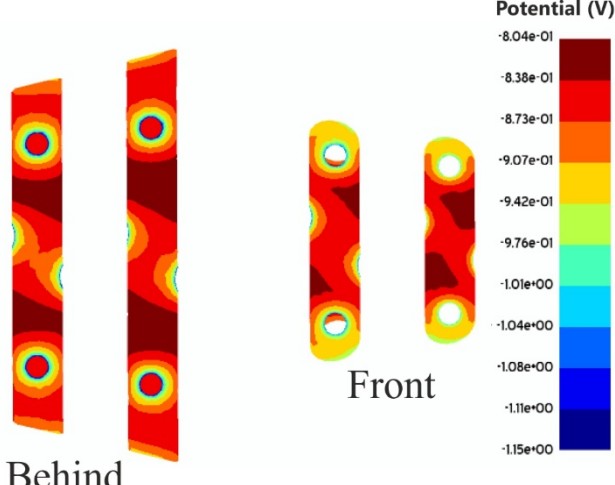

**Figure 12.** Potential distribution φ (V vs Ag/AgCl/seawater) at the steel tubes of the thrusters.

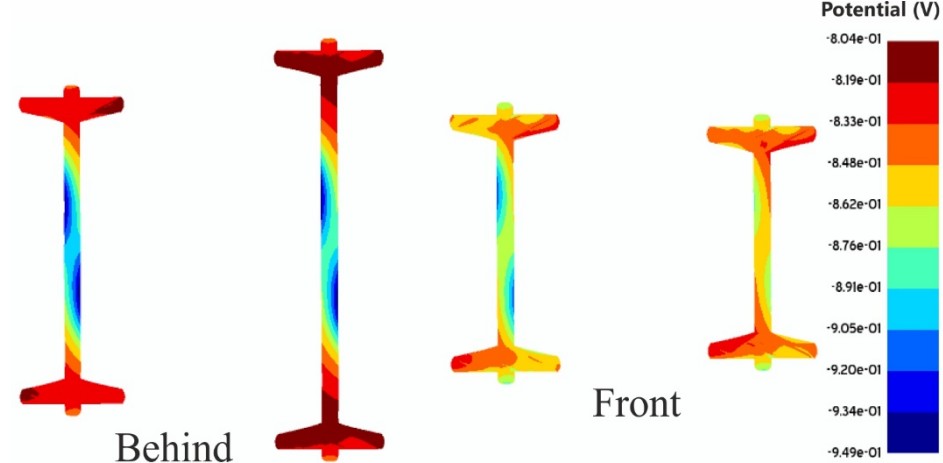

**Figure 13.** Potential distribution φ (V vs Ag/AgCl/seawater) at the steel maneuvering propellers of the thrusters.

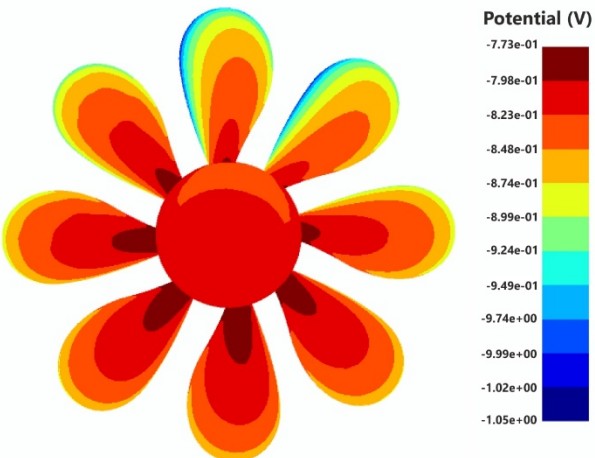

**Figure 14.** Potential distribution φ (V vs Ag/AgCl/seawater) at the NAB main propeller.

From Figures 9 and 10, one can observe that the electric potential at the hull varies in the range of −0.801 and −1.53 V (vs. Ag/AgCl/seawater RE). Similarly, Figure 11 reveals that, the potential at the rudder is in the range of −0.812 and −0.865 V, and Figures 12 and 13 show that at the tubes and maneuvering propellers of the thrusters, the potential varies in the ranges of −0.804 and −1.15 V and

−0.804 and −0.949 V, respectively. Finally, from Figure 14, one can observe that the electric potential at the NAB main propeller varies in the range of −0.773 and −1.05 V (vs. Ag/AgCl/seawater RE).

According to DNV [23], the potential of a steel structure, immersed in seawater, must be in the range of −0.8 V to −1.15 V (vs. Ag/AgCl/seawater RE). The minimum protection potential of −0.8 V corresponds to a shift by 0.142 V in the negative direction of the free corrosion potential of steel in seawater (Equation (14)). For less negative potentials than −0.8 V, a steel structure is under-protected, while for more negative potentials than −1.15 V, it is overprotected, where disbonding or blistering of the coating, due to extreme hydrogen generation may occur. However, most of the new marine coating systems can resist cathodic disbonding to much more negative potential values than −1.15 V, even to a potential of −2 V [1].

In conclusion, the contour plots of Figures 9–13 reveal that on all immersed steel surfaces of the ship, the electric potential is more negative than −0.8 V (vs. Ag/AgCl/seawater RE), and thus, all the exposed steel surfaces are cathodically protected. Furthermore, the most negative electric potentials at the rudder, the maneuvering propellers, and the tubes are −0.865 V, −0.949 V, and −1.15 V, respectively, and thus, no overprotection occurs, even if the more restricted limit of −0.15 V is adopted. However, the value −1.53 V exhibited at the hull is more negative than the overprotection limit of −1.15 V introduced by DNV [23] and less negative, than the corresponding one of −2 V adopted by Lee and Lim [1]. Based on our experience, we consider that no overprotection occurred due to the used marine epoxy coating by the proposed ICCP system. I the above-mentioned overprotection limits are adopted, the area of the used electrical insulators must be reconsidered accordingly.

In modern ICCP systems, the delivered current is adjusted automatically. The potential is measured in real-time at appropriate selected locations, utilizing mounded reference electrodes (REs) as sensors, and the ICCP's control unit regulates the current output to maintain the potential at the REs at a predetermined level. In the proposed six-zone ICCP system, one RE sensor is used per zone, while a preset potential level of −850 mV (vs. Ag/AgCl/seawater RE) has been adopted for all zones. Based on the analysis results, the appropriately selected placements are shown in Figure 15.

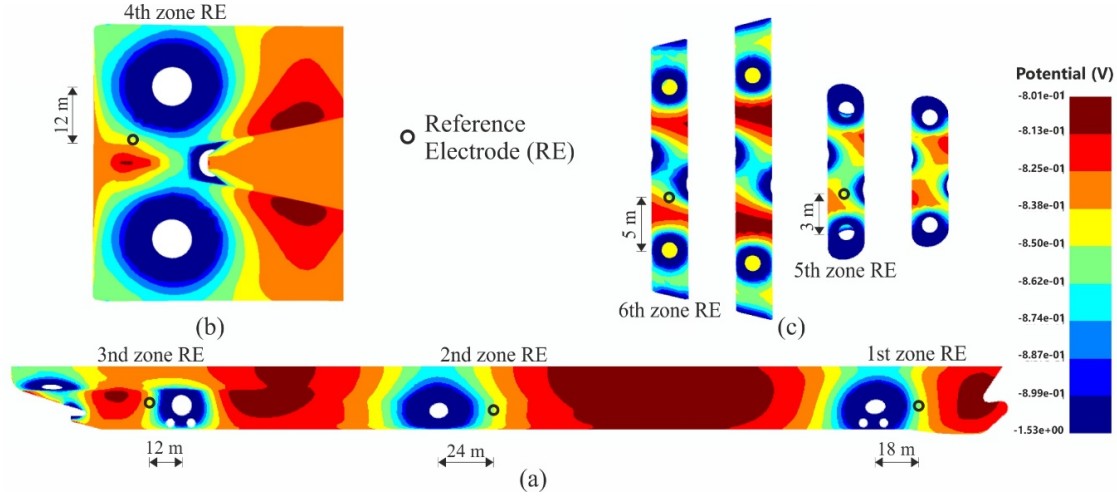

**Figure 15.** Reference electrode (RE) locations for each zone; (**a**) first, second, and third zones; (**b**) fourth zone, and (**c**) fifth and sixth zones.

## 4. Conclusions

In the present work, an ICCP system for an 18270 TEU container ship (length of 399 m, total immersed surface of the vessel is 34,116 m$^2$) was designed, with the aid of an ACA/BEM code, utilizing a detailed large-scale model. The problem is modelled, taking into consideration the exact geometry, the progressive damage of the coating system, and the dynamic state during the cruise of the ship.

In performing an extensive parametric study, a six-zone ICCP system was proposed, consisting of 10 anodes at the hull and four identical anodes at each of four thrusters and a total current demand of 2643 A. The portion of the consumed current is 22.8%at the stern areas, 70.6% at the remaining hull and 6.6% at the thrusters.

According to the obtained numerical results, the exhibited electric potentials are more negative than the minimum protection level of −0.8 V (vs. Ag/AgCl/seawater RE). Thus, all the exposed steel surfaces of the ship are cathodically protected by the proposed ICCP system. However, the most negative value of −1.53 V appeared close to the insulated areas and is more negative than the overprotection limit of −1.15 V introduced by DNV [23] but less negative than the corresponding one of −2 V adopted by Lee and Lim [1]. If the above-mentioned overprotection limits are adopted, the area of the used electrical insulators must be reconsidered accordingly.

**Author Contributions:** Conceptualization, S.V.T. and D.P.; methodology, D.T.K., D.C.R., T.V.G., and S.V.T.; software, D.C.R. and T.V.G.; investigation, D.T.K. and S.V.T.; data curation, D.T.K.; writing—original draft preparation, D.T.K. and S.V.T.; writing—review and editing, D.P.; supervision, D.P. All authors have read and agreed to the published version of the manuscript.

**Funding:** This research was partially funded by "Andreas Mentzelopoulos Scholarships for the University of Patras" (Grant No. 3372). Furthermore, this research has been co-financed by the European Union and Greek National Funds through the Regional Operational Program "Western Greece 2014–2020" (Grant No. 80763), under the call "Regional research and innovation strategies for smart specialization (RIS3) in Information Technology and Telecommunications" (Project: 5038699 entitled "Innovative and Qualitative Software Package for Modeling Industrial Problems in Electromagnetics –IQPATRAS").

**Conflicts of Interest:** The authors declare no conflict of interest.

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
