# Peer review of "Cathodic Protection of A Container Ship Using A Detailed BEM Model"

_jmse, doi:10.3390/jmse8050359_

Round 1

Reviewer 1 Report

Manuscript ref:  JMSE-790546

Dear Editor,

This paper deals the “ICCP system design of a container ship using a large-scale detailed ACA/BEM model” by Dimitrios Kalovelonis et al. is well presented, very scientific and contributes to knowledge immensely. The manuscript is sufficiently detailed and suitable for publication in Journal of Marine Science and Engineering. I strongly recommend publication after Minor Revision.

Points to be corrected:

  1. The title “ICCP system design of a container ship using a large-2 scale detailed ACA/BEM model tle” should not contain less common acronyms (ICCP, ACA/BEM model). Your title distract readers. A good title should contain the fewest possible words that adequately describe the contents of a paper. Please revise the title.
  2. In the title also, what do you mean by the sentence “tle”? Please revise the title.
  3. The Abstract should explain clearly what has done and what are the main findings. The Abstract section should comprise a brief factual account of the contents of the paper, with emphasis on new information. Please to review.
  4. Please revise the Keywords section by reviewing: 1- “Shipboard ICCP system” by “ICCP system”, 2- ”boundary element method, BEM” by 3- ”boundary element method (BEM)” and “2.2 ACA/BEM technology” by “2.2 ACA/BEM method”
  5. Please revise section :
    1. Section “ 2. Governing equations of an ICCP problem for immersed structures in semi-infinite electrolytes and ACA/BEM technology” should be “2. Governing equations of an ICCP problem for immersed structures in semi-infinite electrolytes and ACA/BEM technology”.
    2. Section “3 BEM model” should be “3.4 BEM model”
    3. Section “4 Numerical results and discussion” should be “3.5 Numerical results and discussion”
  6. Equations (4 and 6) should be revised you can use Microsoft Word mathematical equation editor.
  7. The references should identify [4-21] not [4÷21]. Please revise.

Reviewer 2 Report

The authors present an interesting work dealing with the designing of impressed current cathodic protection system of a large container ship using the boundary element method, accelerated by adaptive cross approximation technique. ACA/BEM technology is well described, as well as the design of ICCP system of a ship.

Manuscript has merit, but there are some mistakes which should be corrected.

My comments and suggestions:  

  1. Title, line 3:
    Please review the title (please delete "tle" at the end of the sentence)

  2. Caption of Figure 2, line 166: 
    This is not a view from the front, but the view from the side. 
  3. Figure 3, page 6:
    caption of Fig.3(a) - This is not a top view, but a bottom view.
    caption of Fig.3(b) - This is not a front view, but a side view.

  4. line 204:
    Please review the value of icorr of NAB.
    Since the Nickel-Aluminium-Bronze alloy exhibits excellent corrosion resistance, appearing corrosion rates of 0.015÷0.05 mm/year and consequently no protection is needed, why the corrosion current value is so high, i.e. icorr = 100 mA/m2 ?
    For comparison: the authors adopted icorr only 1.63 mA/m2 for coated steel surfaces.
  5. line 244:
    Please correct the phrase "cathodic desponding".
    Should be: cathodic disbonding.
  6. line 326:
    Please correct the Figure number in the text.
    Should be: "are shown in Figure 15" (instead of Figure 16).
  7. line 408:
    Please correct the description of the standard in the References: the October 2010 edition of DNV-RP-B401 is superseded by the DNVGL-RP-B401 Edition June 2017.

Reviewer 3 Report

This is an interesting paper. I am sure the reader of the journal would benefit from it. However, please reconsider the paper title. Make it easier for the reader to understand the title. If you make it too specific the general reader might not be able to understand the title. There are several places where I thought that the English was unclear. Please reconsider the choice of words and check the grammar. Some other suggestion are given below:

Equation 4- Not legible. Has some random characters in the pdf file

Equation 6- not legible. Has some random characters in the pdf file

Figure 2: Side view or elevation of the ship, not the front view

Line 161: "However, in a galvanic cell, 161 NAB functions as a current sink because of its high electric conductivity property." I am not sure what the authors meant by this statement. Did they mean that NAB acts as a cathode?

Line 224: "In order to avoid cathodic desponding..." Not sure what this means. Please check.
